# The Structural Relationship on Nostalgia Recognition Effect, Attachment, Resilience, and Psychological Well-Being of Dance for All Participants during the COVID-19 Pandemic

**DOI:** 10.3390/healthcare10091793

**Published:** 2022-09-17

**Authors:** Sun-Young Lim, Hyun-Joo Min, Yong-Hwan Kim

**Affiliations:** Department of Physical Education, Gangneung-Wonju National University, Gangneung 25457, Korea

**Keywords:** COVID-19 pandemic, nostalgia recognition effect, attachment, resilience, psychological well-being, dance for all

## Abstract

Background and objectives: During the current era, the world is experiencing economic and mental depression due to COVID-19. In this context, this study empirically analyzed the relationship between the nostalgia recognition effect, a unique human emotion that can improve emotional comfort and stability, and physical activity that can effectively benefit human health; Methods: 550 “dance for all” participants who joined public sports facilities, private gymnasiums, cultural centers, and dance for all clubs in South Korea. Data analysis was performed on Windows PC/SPSS 26.0 and AMOS 24.0 ver. frequency analysis, correlation analysis, confirmatory factor analysis and structural equation modeling were used to analyze the survey results; Results: First, all sub-factors of the nostalgia recognition effect of dance for all participants has a statistically significant (*p* < 0.001) effect on attachment. Second, attachment has a statistically significant (*p* < 0.001) effect on resilience. Third, attachment does not have a statistically significant (*p* < 0.05) effect on psychological well-being of dance for all participants; Conclusions: These research results are prepared effective operating strategies and plans for the era of “post-corona” and “with corona” in the field of human health and dance for all.

## 1. Introduction

After the COVID-19 pandemic struck the world in December 2019, everyday life significantly changed. We sometimes remember beautiful and happy times before the COVID-19 outbreak. The feeling of wanting to return to daily life before COVID-19 is “nostalgia”. Nostalgia is an emotion that provides psychological stability and comfort to those living in a modern society that is politically, economically, and socially uncertain and is also experiencing the crisis of economic growth caused by COVID-19. The more unstable the society, the stronger the desire to recall old memories and return to that previous era for peace of mind, and the desire to avoid reality further reinforces nostalgia [1]. In other words, the recognition of nostalgia serves to stabilize people’s anxiety psychology. The impact of COVID-19 on society in general is enormous, but the psychological impact of the pandemic stands out. COVID-19 is causing a mental health pandemic, where the number of people experiencing mental pain such as anxiety and depression because of social isolation and economic difficulties is continuously increasing [2]. Previous studies have been conducted in related fields, such as studies on baseball fans in Japan and the United States. These studies have shown that nostalgia plays a significant role in the pleasure level spectators feel when consuming sports [3] and plays a significant role in the mechanism of nostalgia formed in childhood and adolescence. These studies indicate that nostalgia acts as a powerful motivator and influences the formation of consumption patterns or product attitudes [4]. These findings suggest that if attachment to a specific product in the past leads to continued purchase after adulthood, it can positively affect not only consumer attitude but also continued purchase intention [5]. 

As such, the perception of nostalgia has a psychological effect and affects consumption behavior. Nostalgia can be considered a kind of “preferential tendency formed through past memories” because it is inevitably accompanied by experiences related to products such as music, movies, fashion, and objects that a person has in the past. In other words, nostalgia perception is closely related to social experience because it is formed by the five human senses (sight, hearing, touch, taste, and smell). In experiential consumption, the positive effect of nostalgia is higher than that of material consumption because the experience itself becomes the aim [6]. In this context, dance for all (DFA) that forms a relationship through experiential consumption of physical activity is easy to apply to nostalgia recognition. DFA is a physical activity that aims to improve welfare by enhancing individual health, aesthetic education, and quality of life through the concepts of “social dance” and “popular dance”. It is also a useful leisure activity that can easily be enjoyed by people of all ages, and is a source of healthy physical training, beauty enhancement, confidence, self-regard, and authentication of others, and is highly effective in relieving the stress caused by COVID-19 [7]. However, due to the prolonged COVID-19 pandemic, national quarantine policies such as social distancing and restrictions on the use of multiple facilities continue to prevent face-to-face activities, the participants’ daily physical dance experiences were curtailed. As a result, participants’ awareness of nostalgia for their participation in pre-COVID-19 DFA increased. Nevertheless, the field of DFA will struggle if an appropriate alternative cannot be found that can satisfy the participants’ desire for physical activity. 

Therefore, this study’s novelty is that it provides practical implications for the application of nostalgia as an alternative to satisfying the desire to participate in physical activities such as DFA during the COVID-19 era. To do so, it explored Therefore, this study explored the structural relationship between nostalgia perception and attachment to DFA, resilience, and psychological well-being (PWB) among dance for all participants (DFAP). This will be of significant help in understanding how the psychology and behavior of DFAP has changed due to COVID-19. In addition, there is an important academic basis for establishing strategies that can revitalize the field of DFA to prepare for “post COVID-19” and “with COVID-19” eras, as well as presenting an efficient operational plan to minimize abandonment or departure from participation in dance due to the prolonged COVID-19 situation.

## 2. Conceptual Background and Hypothesis Setting

### 2.1. The Relationship between NRE (Nostalgia Recognition Effect) and Attachment

The idea of attachment was developed to explain relationships that include a desire to be close to a specific object (including a group member), based on emotional and psychological bonding [8]. It shares common features with the recognition of nostalgia in terms of the emotional bond that can be felt through continuous interaction with a specific object [9]. Thus, it was confirmed that nostalgia recognition allows a positive view of the self through the formation of attachment to a specific object and a strong social bond from being united with others in social relationships [10].

### 2.2. The Relationship between Attachment and Resilience

Resilience is described as emotional vigor, or an individual’s positive ability to mediate the negative effects of stress and promote adaptation, or as a dynamic process in which an individual adapts and copes appropriately when faced with significant and threatening adversity [11]. In other words, resilience acts as a defense mechanism that can maintain psychological well-being in the face of adversity by understanding resilience at the immune system level [12]. Resilience is determined by genetic factors, but is also affected by the passage of time, the individual’s environment, culture, education, and individual efforts [13]. Previous research examined how resilience can be further improved by participating in sports with attachment and showed a causal relationship between place attachment and resilience [14]. 

### 2.3. The Relationship Attachment and PWB (Psychological Well-Being) 

PWB is a factor in the human pursuit of happiness, and it indicates a happy psychological state that enables a positive life. Therefore, PWB is an individual’s subjective evaluation of happiness and a happy life. As it is created by positive emotions and helps a person lead a receptive and positive life, PWB can be used to measure an individual’s quality of life [15]. Related studies on PWB and attachment have confirmed that there is a significant relationship for adolescents, college students, and adults. The degree of secure attachment among maladjusted adolescents was lower than that of general adolescents [16]; studies have also shown that college students with stable attachment have confidence, good interpersonal skills, and form long-term close relationships with others [17]. 

### 2.4. Hypothesis Setting and Research Model

Based on the results of previous studies, the following hypotheses and models were established (Figure 1).

**H1:** 
*NRE of DFAP will have a significant effect on attachment.*


**H1-1:** 
*The positive influence (PI) of DFAP will have a significant effect on attachment.*


**H1-2:** 
*The self-positivity (SP) of DFAP will have a significant effect on attachment.*


**H1-3:** 
*The social connectedness (SC) of DFAP will have a significant effect on attachment.*


**H2:** 
*DFAP’s attachment will have a significant effect on their resilience.*


**H3:** 
*DFAP’s attachment will have a significant effect on their PWB.*


## 3. Methods

### 3.1. Participants

In 2020, the Ministry of Culture, Sports and Tourism of South Korea, Seoul published the National Sports for all Participation Status which stated that Seoul and the Gyeonggi-do region had the highest number of sports club members out of 17 cities and provinces nationwide [18]. Based on this, 550 DFAP who had joined public sports facilities, private academies, and clubs in the city or region area were judged to reflect the various personal characteristics of the research participants and were selected as the sample group. The sample size is the number of completion responses that you receive in the survey, which is called a sample because it represents only a part of a group of people, and you want to know their opinions or behavior [19]. For example, you can use a random sample that selects respondents entirely by chance from a large population as one sampling method (Figure 2).

A survey designed to measure the NME of DFAP was conducted. Participants were limited to those who had begun participating in DFA by at least March 2019 (one year prior to the beginning of the COVID-19 pandemic and continuously participated through January 2022. To identify participants, the purposive sampling method used among non-probability sampling (random sample), and the participants used the self-administration method where they completed the questionnaire and then handed it directly back to the researcher. To increase the understanding and reliability of the survey, researchers visited the site directly and conducted a face-to-face survey in which the questionnaire was distributed and retrieved. All participants provided informed consent. This research was approved by the Institutional Ethics Review Committee of Gangneung-Wonju National University and complied with research ethics. Questionnaires was distributed to a total of 550 participants, and 527 questionnaires were selected and analyzed as final valid samples, excluding 23 that provided incomplete answers such as missing entries, double entries, and biased entries (Table 1).

### 3.2. Measurement Tool

This study used a questionnaire that consisted of questions used and verified through previous studies. The questions for measuring the NRE of DFAP were based on the scale developed by Hepper et al. [20] and were modified and supplemented to match the purpose and subject of this research. The detailed measurement questions consisted of four items about positive influence (PI), four items about self-regard (SR), and three items about social connectedness (SC). The questions that measured attachment for DFAP consisted of three items defined by Ryan and Trauer [21] to meet the purpose and subject of this research. The questions to measure COVID-19 resilience and PWB were composed of five items developed by Luthans [22] and two items based on Ryff’s scale, modified and supplemented to match the purpose and subject of this research. All questions, except for the Demographic characteristic of participants, were measured using a 7-point Likert scale (1 = not at all, 7 = strongly agree). 

### 3.3. Validity and Reliability of Measuring

A group of five experts, consisting of a professor in Sports Pedagogics, DFA (dance for all) professor, Sport Pedagogics Ph.D., and DFA Ph.D, verified the validity of the questionnaire’s contents. The content validity verification is conducted to confirm the appropriateness and representativeness of the question, and to verify whether each question is appropriate for the evaluation of the purpose as well as whether the content of the question faithfully represents the content to be measured [20]. In addition, a confirmation factor analysis (CFA) was performed to present the contents of the questionnaire and to verify the discriminant validity thereof (Table 2) using the AMOS program, which was also used to implement the structural equation model. The number of factors and the measurement variables constituting them are designated prior to the analysis. Therefore, in the CFA, it was assumed that specific measurement variables are necessarily affected only by related factors and are not related to other factors based on strong theoretical background or previous research. In other words, confirmatory factor analysis can verify the discriminant validity that the correlation with other variables except for the measurement variable should be low [21].

According to Bagozzi and Dholakia [22], the best model was evaluated when CFI, NFI, and TLI were 0.8~0.9 or more, and RMR and RMSEA were 0.05 or 0.08 or less, respectively. As a result of conducting confirmatory factor analysis based on this rationale, the model fit of this research was *χ*^²^ = 853.732, *df* = 238, CFI = 0.900, NFI = 0.922, TLI = 0.927, RMR = 0.058, and RMSEA = 0.067 which satisfies the acceptance level suggested by Bagozzi and Dholakia and indicating that it is a good model. Also, the construct reliability of all variables was 0.752~0.838, and the AVE was 0.881~0.967, indicating that the fit criteria suggested were (eigen value > 0.5, CR > 0.7, AVE > 0.5). Each variable was found to have concentrated validity by satisfying the values. Kim [20] explained that reliability is acceptable if the alpha coefficient is 0.5 or more when the reliability test conducted for all questions. After conducting the internal consistency reliability analysis method with Cronbach’s α, the value of Cronbach’s α 0.881–0.904 with relatively high reliability.

### 3.4. Data Analysis Process

The questionnaire used for the final analysis was the result of a data analysis using Windows PC/SPSS 26.0 ver. and AMOS 24.0 ver. after coding and error reviews. First, demographic characteristic of the research participants were analyzed using a frequency analysis. Second, CFA (confirmatory factor analysis) was performed to verify all factors, and reliability verified by calculating the Cronbach’s α coefficient to ensure internal consistency reliability. Third, correlation analysis was performed to analyze the relationship between variables, and structural equation modeling (SEM) was performed to derive a structural model.

## 4. Results

### 4.1. Correlation Analysis

As a result of the correlation analysis between each variable, it was found that there was no multicollinearity problem because no variable showed a correlation of 0.8 or higher in the range of the correlation coefficient value [23] of 0.437 to 0.786 (Table 3).

### 4.2. Model Verification

The results of the analysis verified the suitability of the structural model established in this research: *χ*² = 1824.270 (*df* = 181, *p* = 0.000), CFI = 0.964, NFI = 0.949, TLI = 0.950, RMR = 0.047, and RMSEA = 0.065. According to Kline (1998), when the indicators of CFI, NFI, and TLI, which generally evaluate the overall fit of a model, are above 0.8 to 0.9, RMR and RMSEA are evaluated as a good model when they are less 0.8 [24]. Therefore, it was confirmed that this research model explains the research hypothesis and empirical dataset as a suitable model for adoption relatively well (Table 4).

### 4.3. Hypothesis Testing

Based on the results of the testing analysis, H1-1 (*β* = 0.308, *t* = 10.122), H1-2 (*β* = 0.448, *t* = 13.462), and H1-3 (*β* = 0.484, *t* = 9.716) were accepted. H2 (*β* = 0.308, *t* = 10.122) was also accepted, while H3 (*β* = 0.059, *t* = 1.019) was rejected (Table 5).

## 5. Discussion

These results propose a discussion based on the results of analyzing the structural relationship between the NRE, attachment, resilience, and PWB of the DFAP during the COVID-19 pandemic. First, H1-1, that PI of DFAP will have a significant effect on attachment, was accepted. These results are similar to those in prior research which indicated that form attachments when recognizing that nostalgia’s emotions have a positive effect on humans and have a positive effect on changes in human behavior [25]. The cognitive effect of nostalgia is a concept that includes both positive and negative aspects. However, when NRE emotions positively recognized, they can be evaluated by highlighting their positive function in human attachment, attitude, and behavior change [26]. According to the emotion critic’s theory, various emotions can occur depending on which events or issues are evaluated and interpreted; negative emotions lead to pessimistic evaluations or judgments, and positive moods lead to favorable attitudes or positive judgments [27]. In this context, Cheung et al. [28] and Pascal et al. [29] suggest that when people experience and are stimulated by nostalgia, they form positive feelings of warmth, pleasure, and attachment, and have an active attitude toward experiencing and choosing the presence of nostalgia. Accordingly, if the COVID-19 restrictions cause the social atmosphere to be stagnant, participants who positively recognize their nostalgia for DFA will become more attached to DFA. Moreover, when the social atmosphere is stagnant in the COVID-19 situation, DFAPs recover their emotions through positive recognition of nostalgia for DFA, thereby provoking stronger attachment to DFA. 

Second, H1-2, that SP of DFAP will have a significant effect on attachment, was accepted. Wildschut et al. [30] analyzed the content of experiences that evoke nostalgia and reported that important episodes in life induce nostalgia, revealing that SP promoted through nostalgia recognition. Coopersmith [31] and Vess et al. [32], support the results of this research by indicating that SP with high nostalgia recognition forms and induces attachment. 

Third, H1-3, that SC of DFAP will have a significant effect on attachment, was accepted. SC is a response to relationships with others [33] and is a mental expression of past experiences in interpersonal relationships as well as a sense of psychological belonging. If past experiences act as positive emotional factors, social solidarity with others who have shared past experiences can also be viewed as nostalgia recognition. Ashforth and Mael [34] revealed that emotional attachment is induced when nostalgia from empathy for the values of the past acts as a tool to express the meaning of the relationship to which one belongs by recalling past experiences. In addition, Kwon [35] stated that current loneliness reminds us of objects that can reflect past social bonds of nostalgia, makes us think that we belong to those objects, and forms a strong attachment. These results show that the effects of nostalgia recognition and attachment are mutually influential. Through the nostalgia recognition effect of positively recognizing experiences, such as memories and videos of participating in DFA, indicated that DFAP received comfort that mitigated their anxiety related to COVID-19, which provoked a strong attachment to DFA. Currently, the livelihood of the dance sector is threatened by the difficulty caused by restricted physical leisure activities because of COVID-19. As we have entered the era of “with corona” and “post-corona”, we can no longer wait for the end of COVID-19. We need to move away from the idea that DFA is field art and provide an alternative appropriate for the current situation. Therefore, the most urgent issue in the DFA field is establishing a systematic online platform so that participation in DFA can continue without face-to-face contact. New online platforms are being created, but the current online education platforms for direct participants in DFA are of poor quality. Online platforms actively utilize various contents that can stimulate nostalgia, remote program operation, interactive dance platform, and a global OTT platform that enables two-way communication in a virtual format. In addition, the DFA field also needs to strengthen the attachment of DFAP as it increases the competitiveness of DFA itself. Attachment includes not only interpersonal attachment but also place attachment, where participants feel a sense of belonging through psychological, emotional, and behavioral connections to a specific place [36]. Therefore, even in a face-to-face situation where life dance is limited, it is necessary to select work concepts, music, and costumes that stimulate nostalgia in the DFAP and induce attachment to DFA. Also, if an environment in which nostalgia can be recognized using space, such as interior decoration and lighting, is created, it can make DFAP want to continuously participate in DFA. In addition, events such as daily dance participation videos and presentations, DFA club meetings, and various promotional strategies using SNS can strongly stimulate the participants perception of nostalgia to form an attachment to life dance, which can lead to continuous participation. Events such as DFA participation videos, performance, DFA club meetings, and various promotional strategies using SNS strongly stimulate participants perception of nostalgia to form an attachment to DFA, which can lead to continuous participation. 

Fourth, H2, that attachment of DFAP will have a significant effect on resilience, was accepted. Lee and Kim [37] reported that the more frequently amateur golfers participate in golf, the higher their resilience. Choi and Kim [38] found that the higher the golf participants’ attachment to the golf course, the stronger the positive effect on their resilience, which is consistent with the results of this study. 

Finally, H3, that attachment of DFAP will have a significant effect on PWB, was rejected. This is contrary to studies by Armsden and Greenberg [39], which revealed that university students with stable attachment had a higher ability to adapt to life than university students with unstable attachment, and Stipek et al. [40] reported that attachment-forming adolescents could control anxiety easily when stressed. These results suggest that DFAP’s attachment to DFA has a positive effect on resilience to overcome COVID-19 depression. Although it had a positive effect on resilience in the microscopic aspect, it did not affect PWB in the macroscopic aspect. This may reflect the special situation of the COVID-19 pandemic, and a better, step-by-step daily recovery plan is required to improve the quality of life and satisfaction with COVID-19 policies and practices by national and local governments.

## 6. Limitations of the Research

First, since this research was limited the population of DFAP in Seoul and the Gyeonggi-do region of South Korea, the problem of representativeness of the sample may be raised in generalizing the results presented in the study. Therefore, in follow-up research, expanding the sampling area and participants for comparison and analysis is necessary.

Second, participation in DFA as a physical activity is limited to dance art (Korean dance, ballet, modern) and dance sports, as well as line and K-pop dancing, making it difficult to generalize the results presented in this research to the entire field of DFA. Therefore, in follow-up research, it is necessary to select more diverse DFA fields and conduct specific research considering the characteristics of each field of DFA.

Finally, in this research, the NRE, attachment, resilience, and PWB variables did not extend to an analysis of the contribution according to demographic characteristic of participants (gender, age, field, experience, frequency, and level of DFAP’s participation). Therefore, in follow-up research, an in-depth analysis and examination of the influence relationship according to the demographic characteristics of participants should be conducted.

## 7. Conclusions

Recently, along with the economic difficulties caused by COVID-19, physical, mental, and social health problems have emerged worldwide. Therefore, in this research, the effect of nostalgia perception was set as an important factor in defending against anxiety caused by COVID-19 in DFAP. Based on the results and discussion of this research, the following conclusions were drawn. First, because of testing the hypothesis that NRE will have a significant effect on attachment, all sub-factors (PI, SP, SC) have a significant effect on attachment. Second, testing showed that DFAP attachment to DFA has a significant effect on the resilience of COVID-19. Third, the attachment of DFAP to DFA was shown to haven’t a significant effect on PWB of COVID-19. Finally, the moderating effect on gender to all sub-factors of the NRE and attachment have a statistically significant difference. These results can be used as foundation to create a knowledge system that establishes an efficient operational strategy for overall dance in life, which was inadequate to cope with the COVID-19 situation.

## Figures and Tables

**Figure 1 healthcare-10-01793-f001:**
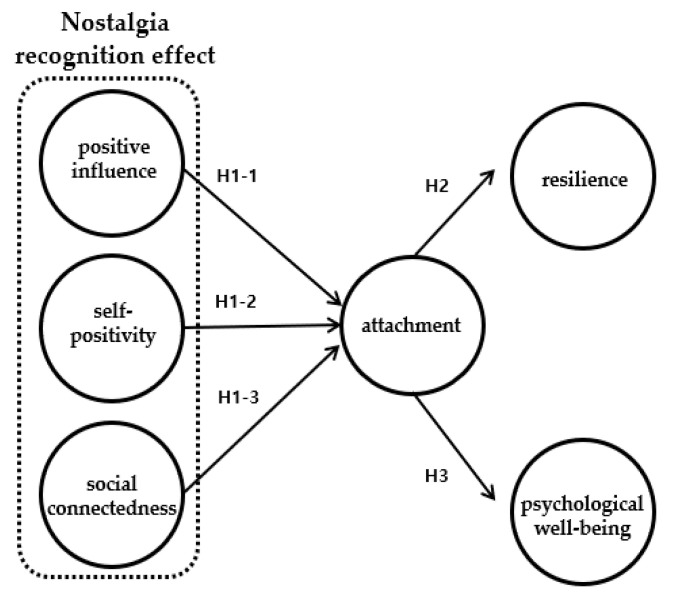
Research model.

**Figure 2 healthcare-10-01793-f002:**
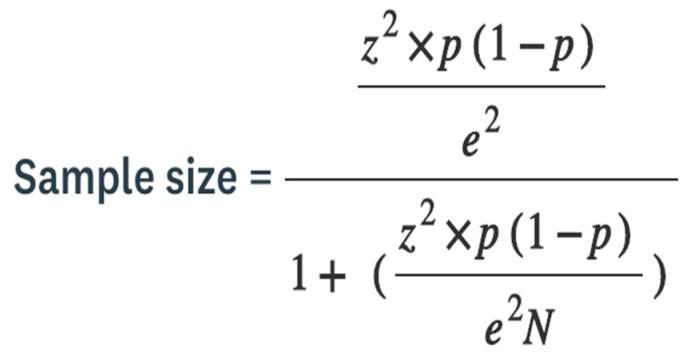
Sample size calculation equation.

**Table 1 healthcare-10-01793-t001:** Demographic characteristic of participants.

Variables	Classification	Frequency (N)	Percentage (%)
Gender	Male Female	221 306	41.9 58.1
Age	10 s 20 s 30 s Over 40	129 166 124 108	24.5 31.5 23.5 20.5
Participation field	Dance (Korean dance, ballet, modern dance) Dance sports Line dance K-pop dance	90 138 120 179	17.1 26.2 22.8 33.9
Participation experience	2 yrs–less than 3 yrs 3 yrs–less than 4 yrs More than 4 yrs	291 105 131	55.2 19.9 24.9
Participation frequency	Every week Every other week Every three weeks Once a month	173 215 77 62	32.8 40.8 14.6 11.8
Level of physical participation	High Middle Low	192 248 87	36.4 47.1 16.5
Total	527	100	100

**Table 2 healthcare-10-01793-t002:** Confirmatory factor analysis.

Variables	Factors	*SC*	*SE*	*t*	C.R	AVE	Cronbach’s α
Positive Influence (PI)	Participating in DFA before COVID-19 made me happy.	0.868	-	-	0.752	0.881	0.894
	Participating in DFA before COVID-19 made me feel better.	0.878	0.036	28.42			
	Participation in DFA before COVID-19 made me active	0.861	0.037	27.40			
	Participating in DFA before COVID-19 made me feel calm.	0.674	0.038	18.86			
Self-positivity (SP)	Participating in DFA before COVID-19 made me discover a good side of myself.	0.750	-	-	0.757	0.883	0.923
	Participating in DFA before COVID-19 made me feel more valuable.	0.793	0.045	23.44			
	Participating in DFA before COVID-19 made me a better person.	0.945	0.050	25.48			
	Participating in DFA before COVID-19 made me like myself even more.	0.849	0.048	23.87			
Social connectedness (SC)	Participating in DFA before COVID-19 made me feel loved.	0.840	-	-	0.760	0.853	0.859
	Participation in DFA before COVID-19 made me feel connected to the people around me.	0.820	0.052	20.68			
	Participating in pre-COVID-19 DFA made me feel strong and protected.	0.886	0.057	20.59			
Attachment	I felt good when I participated in DFA.	0.881	-	-	0.838	0.894	0.924
	I felt happy when I participated in DFA.	0.938	0.031	34.10			
	I felt calm and comfortable when I participated in DFA.	0.872	0.033	29.47			
Resilience	Even if I face difficulties in my DFA, I can overcome them quickly.	0.671	-	-	0.756	0.904	0.875
	I can withstand difficult situations without any problems.	0.642	0.049	20.10			
	Even if there is a problem in daily life, it will be easily solved feel.	0.835	0.071	17.62			
	My life will turn out the way I want it to.	0.641	0.063	15.58			
	Even if I feel frustrated in my daily life, I can shake it off and recover.	0.761	0.069	16.29			
Psychological well-being (PWB)	I feel that I’m doing my job (responsibility) well.	0.996	-	-	0.837	0.846	0.659
	I feel good at managing personal problems that occur in my day-to-day life.	0.723	0.046	15.72			
*χ*^²^ = 853.732 (*df* = 238, *p* = 0.000), CFI = 0.900, NFI = 0.922, TLI = 0.927, RMR = 0.058, RMSEA = 0.067							

DFA (dance for all); DFAP (dance for all participants).

**Table 3 healthcare-10-01793-t003:** Correlation analysis.

Variables	Positive Influence	Self- Positivity	Social Connectedness	Attachment	Resilience	Psychological Well-Being
Positive influence	1					
Self-positivity	0.786 **	1				
Social connectedness	0.643 **	0.665 **	1			
Attachment	0.684 **	0.676 **	0.719 **	1		
Resilience	0.738 **	0.779 **	0.649 **	0.711 **	1	
Psychological well-being	0.464 **	0.437 **	0.439 **	0.453 **	0.524 **	1

** *p* < 0.01.

**Table 4 healthcare-10-01793-t004:** Fit index of research model.

A Construct	*χ*²	*df*	*p*	CFI	NFI	TLI	RMR	RMSEA
**Acceptance level**	1824.270	181	0.000	0.964	0.949	0.950	0.047	0.065

**Table 5 healthcare-10-01793-t005:** Hypothesis testing result.

H	Path	*SE*	*CR*	*p*	Accept/Reject
H1-1: Positive influence → Attachment	0.308	0.003	10.122	0.000 ***	Accept
H1-2: Self-positivity → Attachment	0.448	0.033	13.462	0.000 ***	Accept
H1-3: Social connectedness → Attachment	0.484	0.050	9.716	0.000 ***	Accept
H2: Attachment → Resilience	0.639	0.044	14.385	0.000 ***	Accept
H3: Attachment → Psychological well-being	0.059	0.058	1.019	0.308	Reject

*** *p* < 0.001.

## Data Availability

Not applicable.

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
