# Peer review of "The Structural Relationship on Nostalgia Recognition Effect, Attachment, Resilience, and Psychological Well-Being of Dance for All Participants during the COVID-19 Pandemic"

_healthcare, 2022, doi:10.3390/healthcare10091793_

Round 1

Reviewer 1 Report

 Thank you for the opportunity to review this manuscript.

1-      In the Conceptual background and hypothesis setting, page 2, lines 92-93. The authors stated that “Previous studies have shown that nostalgia recognition induces attachment behavior and maintains a strong bond with the object of attachment [10]”.

Reference 10 is a single one study, not a systematic review, therefore, the authors need to modify instead of previous studies to a similar study, or a previous study….

2-      The age range of the participants is wide from 10+ to above 40s some confounding variables may contribute to the relationship between nostalgia and attachment to DFA, and resilience in different age groups.

The authors need to debate more on this issue in the results and discussion section to verify the internal validity of this study.  

3-      In the discussion section, the authors stated that “Many studies have reported that emotions play an important role in guiding people's decision-making and behavior [29]”.

As shown in the references, reference #29: Kline R.B.; Software review: software programs for structural equation modeling: Amos, EQS, and LISREL. Journal of Psychoeducational Assessment, 1998, 16, 343-364.

My comment is that it seems that reference # 29 is about software programs for structural equation modeling, and not related to the context in this sentence. Please revise.

Author Response

Dear reviewer A

Thank you very much for reviewing the research. And thank you very much for giving us a chance to revise.

Our researchers very much respect reviewers' decisions and comments. Our researchers are still academically scarce. It is a team that strives to advance through a repeated challenging process of research, submission, and revision.

We tried to revise as much as possible following the reviewer's advice. Nevertheless, more modifications may still be required. We ask for reviewer's understanding. And if our research needs further modifications, we'll work hard again and again.

The revised contents are as follows.

*Please refer to the attached file.

Thank you again.

Reviewer 2 Report

Introduction - it is advisable to write in parentheses the name of the abbreviations used, i.e. BTS, DFA, DFAP etc., not all readers understand what mean these abbreviations.

Also, highlight very clearly in this chapter what was the novelty of this study.

Line 125 - is Fig 1 or ''Figs. 1'', please correct.

Recruitment of participants - were there any people who refused to participate or did not meet all the conditions to be included in this study? if yes, please fill with the necessary information, not only ''23 gave incomplete answers.''

''For the participant sample, the purposive sampling method was used among non-probability sampling, and the participants used the self-administration method where they completed the questionnaire and then handed it directly back to the researcher''....ok, but what questionnaire is it about? please present other information regarding this evaluation tool so that all the readers can understand very clearly the application methodology in this study.

Even if the authors presented information about the measuring instruments used, see lines 158-168, these are insufficient. It is advisable to add as much information as possible for each evaluation tool applied, to present the scoring of each item, etc.

Also, please present what were the limits of this study.

References - please use the abbreviations for each journal, not the full name.

Author Response

Dear reviewer B

Thank you very much for reviewing the research. And thank you very much for giving us a chance to revise.

Our researchers very much respect reviewers' decisions and comments. Our researchers are still academically scarce. It is a team that strives to advance through a repeated challenging process of research, submission, and revision.

We tried to revise as much as possible following the reviewer's advice. Nevertheless, more modifications may still be required. We ask for reviewer's understanding. And if our research needs further modifications, we'll work hard again and again.

The revised contents are as follows.

*Please refer to the attached file.

Thank you again.

Round 2

Reviewer 2 Report

I still did not understand what is it ''BTS'', please provide additional data about this abbreviation. The authors say that it represents "South Korea of ​​K-POP idol group'', I don't understand very clearly how the name in the parenthesis correlates with the abbreviation. What is B, T, S?

From DFA (dance for all) and DFAP (dance for all participants) it is very clear and understandable to the readers, but for BTS.......

Lines 127, 128, 152, 154 - It is Fig., no ''Figs'', right?, you don't correct as I suggested before, please do that.

Author Response

(The authors gave the same response as above.)
